# CircCSDE1 Regulates Proliferation and Differentiation of C2C12 Myoblasts by Sponging miR-21-3p

**DOI:** 10.3390/ijms231912038

**Published:** 2022-10-10

**Authors:** Di Sun, Jiaqi An, Zixu Cui, Jiao Li, Ziwei You, Chang Lu, Yang Yang, Pengfei Gao, Xiaohong Guo, Bugao Li, Chunbo Cai, Guoqing Cao

**Affiliations:** College of Animal Science, Shanxi Agricultural University, No. 1 Mingxian South Road, Taigu 030801, China

**Keywords:** pig, skeletal muscle injury, circCSDE1, C2C12, proliferation, differentiation

## Abstract

The growth and development of skeletal muscle is regulated by many factors, and recent studies have shown that circular RNAs (circRNAs) can participate in this process. The model of porcine skeletal muscle injury was constructed to search for circRNAs that can regulate the growth and development of skeletal muscle in pigs. Using whole-transcriptome sequencing and bioinformatics analysis, a novel circRNA (circCSDE1) was screened out, which is highly expressed in skeletal muscle. Functional studies in C2C12 cells demonstrated that circCSDE1 could promote proliferation and inhibit myoblast differentiation, while opposing changes were observed by circCSDE1 knockdown. A dual-luciferase reporter assay revealed that circCSDE1 directly targeted miR-21-3p to regulate the expression of the downstream target gene (Cyclin-dependent kinase 16, *CDK16*). Moreover, miR-21-3p could inhibit proliferation and promote myoblast differentiation in C2C12 cells, opposite with the effects of circCSDE1. Additionally, the rescue experiments offered further evidence that circCSDE1 and its target, miR-21-3p, work together to regulate myoblast proliferation and differentiation. This study provides a theoretical basis for further understanding the regulatory mechanisms of circRNAs.

## 1. Introduction

Skeletal muscle is a metabolically active tissue distributed throughout the animal body. It represents about 50% of total body weight [1] and plays an important role in maintaining energy homeostasis [2]. The formation of skeletal muscle is a complex process, beginning with the differentiation of mesenchymal stem cells into myoblasts that further differentiate and fuse to form multinucleated myotubes. Then, multiple myotubes fuse to form muscle fibers, which eventually mature to form skeletal muscle [3]. Many well-known transcription factors display regulatory effects on myogenesis: the myocyte enhancer factor 2 (*MEF2*) family [4], myogenic regulatory factors (*MRF*) family [5], paired box 7 (*PAX7*) and paired box 3 (*PAX3*) [6]. Together, these factors cooperatively regulate the controlled process of myoblast proliferation and differentiation. Recent studies further revealed that various non-coding RNA species, including lncRNA [7,8,9], miRNA [10,11,12], and circRNA [13,14,15], participate in the regulation of myogenesis. In particular, circular non-coding RNA molecules (circRNAs) are increasingly recognized as key regulatory factors of various biological processes, but their functional roles in myogenesis remain poorly defined.

CircRNAs are formed by the reverse splicing and cyclization of certain pre-mRNAs with compatible secondary structures [16]. Small circRNAs were first identified in the 1970s in viruses [17], but due to technical limitations at the time, they could not be isolated and studied in depth. This led to circRNA generally being considered as an abnormal shear body with no known biological function [18]. However, recent studies showed that circRNAs can participate in the regulation of myoblast proliferation and differentiation through the competing endogenous RNA (ceRNA) mechanism. For example, circTTN targets miR-432 to alleviate its inhibitory effect on insulin-like growth factor 2 (*Igf2*) expression, thus promoting proliferation and differentiation of skeletal muscle cells in cattle [19]. In pigs, circTUT7 can increase the expression of high mobility group 20B (*Hmg20b*) by sequestering miR-30a-3p, thus promoting transcription of genes related to myogenesis [20]. However, there are few studies on the growth and development of skeletal muscle in pigs, so it is difficult to fully understand the regulatory effects of circRNAs on these processes. We therefore conceived the present study to provide additional insights into the regulatory network of circRNAs affecting skeletal muscle growth and development in pigs.

To identify such circRNAs, a model of pig skeletal muscle injury was constructed. Samples from different treatment groups were subjected to whole-transcriptome sequencing, and many differentially expressed circRNAs were identified by bioinformatics analysis. Among these, circCSDE1 was highly expressed in skeletal muscle and differentially expressed across treatment groups. Further studies showed that circCSDE1 regulates the growth and development of skeletal muscle. This work laid the foundation on the further investigations of circCSDE1 regarding pig skeletal muscle growth and development.

## 2. Results

### 2.1. Overview of Sequencing Sata

The basic information on the sequencing results is presented in Appendix A. In the 90d group, an average of 101 million raw reads was generated, with about 100 million clean reads obtained after quality control. In the MI group, an average of 90.8 million raw reads and 90.2 million clean reads were obtained. The equivalent results were, on average, 98.6 million raw reads and 97.9 million clean reads in the R_10d group. Moreover, the error rate of all 12 sequencing data sets was below 0.03%, with quality scores of Q20 for >97% and Q30 for >92% of bases. These data confirm the authenticity and accuracy of the sequencing results.

### 2.2. Basic Characteristics of circRNAs

A total of 17,532 candidate circRNAs were identified by analysis of the whole-transcriptome sequencing data. Their genomic loci are widely distributed across all chromosomes (Figure 1A), with varied source types: approximately 94.80% exons, 3.50% introns and 1.66% intergenic regions (Figure 1B). The size distribution (Figure 1C) shows that the majority of these circRNAs are around 300 nt in length.

### 2.3. Differentially Expressed circRNAs

Compared to the 90d group, 335 circRNAs were upregulated and 191 circRNAs were downregulated in the MI group (Figure 2A; Appendix A). Compared to the 90d group, 241 circRNAs were upregulated and 165 circRNAs were downregulated in the R_10d group (Figure 2B; Appendix A). Compared to the MI group, 42 circRNAs were upregulated and 43 circRNAs were downregulated in the R_10d group (Figure 2C; Appendix A). The Venn plot (Figure 2D) showed that three circRNAs (circ_0010134, circ_0007453 and circ_0006532) were consistently differentially expressed among the experimental groups.

### 2.4. CircCSDE1 can Stably Form Circular Structures

The circ_0006532 contains 299 bases. The alignment of the sequenced circ_0006532 to the pig genome using Nucleotide BLAST revealed that circ_0006532 fully aligned with the mRNA sequence of porcine cold shock domain-containing E1 (*Csde1*, *XM_003481484.4*). Bases 1–110 and 202–299 of circ_0006532 completely matched the exon 5 and 7 sequences of *Csde1*, respectively. Additionally, the bases 111–201 matched the exon 6 without the first (G) and last (G) base (Figure 2A). So, the circ_0006532 was named circCSDE1. The cyclization site (CAATG) of circCSDE1 contains the first 3 bases (ATG) of exon 5 and the last 2 bases (CA) of exon 7 (Figure 2A). Convergent and divergent primers of circCSDE1 were designed for PCR amplification using cDNA and gDNA as templates, respectively. Both Convergent and divergent primers produced bands when the cDNA was used as the template, but only the convergent primers produced a band when gDNA was used as the template (Figure 3B). The PCR product amplified from cDNA by the divergent primers was then sequenced, revealing the cyclization site sequence CAATG (Figure 3C), consistent with the transcriptome sequencing results. Furthermore, the RNase R digestion assay showed that after exposure to RNase, the amplification of circCSDE1 did not significantly change, whereas the amplification of linear *Csde1* and 18S rRNA significantly decreased (Figure 3D). Taken together, the present results show that circCSDE1 is an endogenous circRNA stably expressed in porcine skeletal muscle.

### 2.5. Analysis of circCSDE1 Expression Patterns

The expression of circCSDE1 in R_10d group was the highest, meanwhile the expression in MI group was also significantly higher than that in the 90d group, which displayed the same trend as the transcriptomic data (Figure 4A). CircCSDE1 was highly expressed in lung, skeletal muscle, and fat (Figure 4B). Compared with the age of 0 days, the expression of circCSDE1 fell slightly at 90 days, but increased about fivefold by 180 days of age (Figure 4C). Subcellular fractionation experiments established that circCSDE1 was mainly localized in the cytoplasm with *U6* and glyceraldehyde-3-phosphate dehydrogenase (*Gapdh*) as nuclear and cytoplasmic controls, respectively (Figure 4D). CircCSDE1 is also expressed in C2C12 cells by porcine divergent primers. During myogenic differentiation of C2C12, the expression of circCSDE1 increased from 0 to 4 days but decreased at 6 days (Figure 4E).

### 2.6. CircCSDE1 Promotes C2C12 Myoblast Proliferation and Inhibits Differentiation

The overexpression vector for circCSDE1 was constructed and transfected into C2C12 cells. The expression of circCSDE1 was increased by threefold (Figure 5A), accompanied with the significant upregulation of proliferation markers proliferating cell nuclear antigen (*Pcna*), cyclin dependent kinase 4 (*Cdk4*) and marker of proliferation *Ki-67* (*Mki67*) (Figure 5B). Consistently, EdU labeling showed a markedly increased population of proliferating cells upon the overexpression of circCSDE1 (Figure 5C), and a significant increase in proliferation rate was also evident by CCK-8 assay (Figure 5D). Then, the transfected C2C12 cells were induced myogenic differentiation. The differentiation markers myogenic differentiation 1 (*Myod1*), myogenin (*Myog*) and myosin heavy chain (*Myhc*) were significantly downregulated with the overexpression of circCSDE1 (Figure 5E). Moreover, circCSDE1 overexpression was associated with the decreased myotube fusion (Figure 5F).

The same results were obtained in parallel experiments using the siRNA knockdown of circCSDE1 (Figure 6A). The expressions of *Pcna*, *CCND1* (*cyclin D1*), *Cdk4* and *Mki67* were significantly decreased upon circCSDE1 depletion (Figure 6B), and a reduced cellular proliferation rate was also detected by the CCK-8 assay (Figure 6C). In addition, the EdU labeling showed a reduction in proliferating cells within the treatment group (Figure 6D). The differentiation potential of circCSDE1-knockdown in C2C12 cells was also assessed. The marker genes *Myod 1*, *Myog*, *Myhc* and *myogenic factor 5* (*MYF5*) were all significantly upregulated (Figure 6E), and the myotube fusion increased (Figure 6F). The above experiments together demonstrated that circCSDE1 functions to promote proliferation and inhibit myogenic differentiation in C2C12 cells.

### 2.7. CircCSDE1 Acts as a ceRNA for miR-21-3p

The miRDB and RNAhybrid was used to constructed the circCSDE1/miRNA/mRNA network map in Cytoscape (Figure 7A). The predicted interactions between circCSDE1, miR-21-3p, and *CDK16* (circled region) attracted our attention, and the expressions of circCSDE1, miR-21-3p, and *CDK16* in the 90d, MI and R_10d groups were analyzed by qPCR (Figure 7B). The expression of circCSDE1 had a negative correlation with miR-21-3p but a positive relationship with *CDK16*, indicating direct interactions between circCSDE1, miR-21-3p, and *CDK16*. The binding site of circCSDE1 with miR-21-3p was predicted by RNAhybrid (Figure 7C), as well as that of *CDK16* with miR-21-3p (Figure 7D). These targeting relationships were verified by dual-luciferase assays. Reporter activity was significantly decreased with the co-transfection of wild type circCSDE1 vector and miR-21-3p mimic (Figure 7E). Similarly, reporter activity was significantly decreased with the co-transfection of wild type *CDK16* vector and miR-21-3p mimic (Figure 7F). Meanwhile, when circCSDE1 was overexpressed in C2C12 cells, the expression of miR-21-3p significantly decreased, along with increased expression of *CDK16* (Figure 7G). In summary, circCSDE1 can inhibit the expression of miR-21-3p, thereby relieving its inhibitory effect on *CDK16*.

### 2.8. MiR-21-3p Inhibits C2C12 Cell Proliferation and Promotes Differentiation

After establishing the targeting relationship of circCSDE1 with miR-21-3p, the function of miR-21-3p was explored in C2C12 cells. With the transfection of miR-21-3p mimic in C2C12 cells, the expression of miR-21-3p showed an approximately tenfold increase (Figure 8A). Alongside this, the expression of the proliferation marker genes *Pcna*, *Cdk4* and *Mki67* was significantly decreased (Figure 8B), and a slower cell proliferation rate was also detected by CCK-8 assay (Figure 8C) and EdU labeling (Figure 8D). To examine the effect of miR-21-3p on differentiation, the C2C12 cells transfected with miR-21-3p mimic were induced for myogenic differentiation. The differentiation markers *Myod 1*, *Myog* and *Myhc* were significantly upregulated (Figure 8E), and the myotube fusion index significantly increased (Figure 8F).

Conversely, after downregulating the expression of miR-21-3p with siRNA (Figure 9A), the proliferation marker genes *Pcna*, *CCND1*, *Cdk4* and *Mki67* were significantly upregulated (Figure 9B). The CCK-8 assay (Figure 9C), and EdU labeling (Figure 9D) revealed a significantly greater number of proliferating cells. When C2C12 cells were induced for differentiation after siRNA knockdown of miR-21-3p, the expression of differentiation markers *Myod 1*, *Myog*, *Myhc* and *Myf5* was significantly decreased (Figure 9E), and the myotube fusion index was significantly decreased (Figure 9F). Using these overexpression and knockdown experiments, miR-21-3p was demonstrated to inhibit proliferation and promote myogenic differentiation in C2C12 cells.

### 2.9. CircCSDE1 Suppresses Myogenic Differentiation by miR-21-3p in C2C12 Cells

To further explore the relationship between circCSDE1 and miR-21-3p, rescue experiments was performed. The circCSDE1 overexpression alleviated the inhibitory effect of miR-21-3p on myogenic differentiation in C2C12 cells (Figure 10A), with the same result observed in myotube formation (Figure 10B). Meanwhile, the circCSDE1 depletion promoted C2C12 cells myogenic differentiation, and this effect was abolished with simultaneous siRNA targeting of miR-21-3p (Figure 10C). These changes were further confirmed by immunofluorescence staining (Figure 10D). Thus, circCSDE1 attenuates the effects of miR-21-3p in myogenic differentiation.

## 3. Discussion

The process of myoblast development is regulated by many factors, among which non-coding RNA plays a very important role. In this study, three circRNAs (circ_0006532, circ_0010134 and circ_0007453) were identified to be differentially expressed in 90d, MI, and R_10d groups. All the three circRNAs have a high expression in porcine skeletal muscle. However, circCSDE1 (circ_0006532) showed the highest expression difference compared to the different groups (90d/R_10d: log_2_Fold = 6.08; MI/R_10d: log_2_Fold = 6.38). Therefore, circCSDE1 was selected as the candidate non-coding RNA regulating the development of skeletal muscle. Then, combined with the sequencing data, we used RNAhybrid to predict the possible ceRNA regulatory network of circCSDE1. Potential target sites of circCSDE1 included miR-10382, miR-22-5p, miR-21-3p, miR-145-5p and miR-885-5p. Only miR-145-5p was reported to inhibit myoblast differentiation [21]. However, the expression of miR-145-5p did not differ among three treatment groups. Then, we focused on the other miRNAs, whose effects on skeletal muscle have not been reported. Compared with the other three miRNAs, miR-21-3p has a wider range of biological functions in animals. A significantly higher expression of miR-21-3p was found in oral squamous cell carcinomas than in normal tissues, and the proliferative capacity of oral cancer cells was also significantly reduced following transfection with miR-21-3p inhibitors [22,23]. MiR-21-3p may promote the proliferation and migration of ovarian cancer cells [24]. Additionally, the miR-21-3p can also promote hepatocellular carcinoma progression via SMAD7 signal pathway [25]. SMAD7 is an important regulator of skeletal muscle development, and the genetic disruption of *Smad7* impairs skeletal muscle regeneration [26]. SMAD7 can also promote skeletal muscle differentiation [27]. Therefore, miR-21-3p was selected as the candidate target site of circCSDE1. The qRT-PCR results showed that there were indeed significant differences in the expression of miR-21-3p in different treatment groups. Additionally, the expression of circCSDE1 and miR-21-3p showed a negative correlation among different treatment groups. Thus, we focused on circCSDE1 and miR-21-3p to verify their regulatory effects on skeletal muscle development.

The source gene of circCSDE1 is *Csde1*, which encodes the RNA-binding protein cold shock domain-containing protein E1 [28], a cytoplasmic factor involved in regulating the translation of a variety of genes, mainly by binding to the ribosome and inhibiting translation [29]. CSDE1 is also a pleiotropic transcriptional regulator that controls a variety of biological processes, including apoptosis, cell differentiation, and cell migration [30,31,32]. In our study, the *Csde1* gene was found to form the circRNA for the first time, which regulates the proliferation and myogenic differentiation. In this study, it was unclear whether the function of circCSDE1 was affected by its source gene *Csde1*, which will be our next research plan.

CircRNAs have a variety of functions, and the most common of which is to act as ceRNA to regulate the expression of downstream target genes. For example, circZfp609 can competitively bind miR-194-5p, countering its inhibitory effect on *BCL2*-associated transcription factor 1 (*Bclaf1*), thereby inhibiting differentiation of myotubes [33]. CircRNAs can also directly bind to Pol II to regulate transcriptional activity [34]. Specifically, EIciRNA binds to the small ribosomal U1snRNP under the action of RNA polymerase II, thereby increasing the transcription efficiency of the parental gene. CircRNAs can also directly form complexes with proteins to regulate the expression of downstream target genes. For example, circCDRlas can directly bind to argonaute (AGO) proteins, thereby regulating the expression changes of target genes [35]. In this study, we found that circCSDE1 can function as ceRNA to regulate the proliferation and myogenic differentiation.

Five miRNAs were predicted to bite the circCSDE1 by RNAhybrid. The downstream target genes of the five miRNAs were predicted by miRDB to construct a complete ceRNA network, showing that several target genes are related to skeletal muscle development, such as aarF domain-containing kinase 2 (*Adck2*), MYC-associated zinc finger protein (*Maz*) and *CDK16*. The *Adck2* is the target gene of miR-145-5p in the ceRNA network of circCSDE1. *Adck*2^+/−^ mice exhibited impaired mitochondrial myopathy in skeletal muscle resulting in lower physical performance, and significant decrease in Coenzyme Q (CoQ) biosynthesis [36], which indicates that the circCSDE1/miR-145-5p/*ADCK2* pathway can regulate the skeletal muscle development by altering mitochondrial activity. The *Maz* is the target gene of miR-10382 in the ceRNA network of circCSDE1. *MAZ* transactivates the muscle creatine kinase (*MCK*) promoter and binds the *MCK* promoter element X (*MPEX*) site in vitro [37]. *MAZ* was shown to bind and transactivate many muscle promoters including myogenin, *Mef2c* and SIX homeobox 4 (*Six4*) in skeletal myocytes [37]. SIX homeobox 1 (*Six1*) and *Six4* expression is required to specifically activate fast-type muscle genes. The absence of *Six1* and *Six4* leads to the development of dorsal myofibers lacking the expression of fast-type muscle genes and mainly expressing a slow-type muscle program [38,39]. Thus, the circCSDE1/miR-10382/*MAZ* pathway can also regulate skeletal muscle development. The *Cdk16* is the target gene of miR-21-3p in the ceRNA network of circCSDE1. *CDK16* exerts promyogenic effects by regulating myoblast migration and fusion during skeletal myogenesis [40]. It is a member of the CDK-family-containing PSTAIRE motifs, which are important for binding to cyclins [41]. The functions of CDK family members are mainly related to cell proliferation [42]. Previous studies found that CDK16 can be mediated by the phosphorylation of protein regulator of cytokinesis (PRC1) to regulate the cell proliferation [43]. Our data show that circCSDE1 alleviates the inhibitory effect of miR-21-3p on its target gene *Cdk16* to promote proliferation in C2C12 cells.

## 4. Materials and Methods

### 4.1. Experimental Animals and Samples

The experimental animals required for this study (Jinfen White pigs) came from the Pingding Huayi pig farm. Eight 90-day-old castrated Jinfen White boars were divided into two groups. One group (*n* = 4) was slaughtered, and the longissimus dorsi muscle was collected (90 d). The other pigs were used to construct the skeletal muscle injury model. Briefly, 1 mL of atropine sulfate (0.5 mg/mL) and 10 mL of cerazine hydrochloride (50 mg/mL) were injected. When the pigs were in comas, 3 mL lidocaine hydrochloride (20 mg/mL) was injected around the longissimus dorsi muscle. Then, a knife wound was made surgically and the longissimus dorsi muscle was collected (MI). Penicillin powder was sprinkled on the wound and 6 mL dexamethasone sodium phosphate (1 mg/mL) was injected. Then, the wound was sewn up. After 10 days, the longissimus dorsi muscle was collected by the same method (R_10 d). The tissue samples of heart, liver, spleen, lung, kidney, fat, and skeletal muscle were obtained at 90 days of age. The longissimus dorsi muscle of 0 d and 180 d Jinfen White pigs were from laboratory storage.

### 4.2. Library Construction and Sequencing

Total RNA was first isolated from muscle tissue samples using TRIzol reagent (Invitrogen, Carlsbad, CA, USA). Then, the ribosomal ribonucleic acids (rRNAs) were depleted, and the remaining RNAs were fragmented to approximately 200 bp using Ribo-Zero kit (Epicentre, Madison, WI, USA). The cDNA library was prepared with the TruSeq Double-Strand Total RNA Library Prep Kit (Illumina, San Diego, CA, USA) using random primers to generate the fragmented RNA for the first-strand cDNA. In the second cDNA strand, the dNTP reagent in the dTTP was replaced by Dutp, resulting in A/U/C/G bases in the second cDNA strand. After end repair and addition of poly A tails, cDNA fragments of 150–200 bases were isolated, and single-stranded cDNA was obtained using uracil azaglycosylase (UNG). Next, PCR amplification was performed to enrich the library. Finally, sequencing was performed on the Illumina Genome Analyzer II System of Beijing Novogene Bioinformatics Technology Co.Ltd, and the paired-end high-throughput sequence was used. Image digital information conversion was performed using the bcl2fastq Conversion Software package provided by Illumina was used to produce Fastq format data. Then, Trimmomatic, CutAdapter and Fastx (http://hannonlab.cshl.edu/fastx_toolkit/ (accessed on 16 August 2022)) software packages were used to remove linker contamination sequences, then low-quality bases were scanned and filtered out to obtain clean data. Hisat2 (http://ccb.jhu.edu/software/hisat2 (accessed on 16 August 2022)) was used to align clean data to the transcriptome, and the output results are displayed in a standard SAM file. The circRNA were detected and identified using find_circ [44] and CIRI2 [45]. Both softwares run with the default parameters, and no special parameters were set. Differential expression analysis of circRNAs was identified using the DESeq R package (1.24.0). The resulting P values were adjusted using the Benjamini and Hochberg’s approach to controlling the false discovery rate. CircRNAs with an adjusted P-value found by DESeq were assigned as differentially expressed. Additionally, padj<0.05 was regarded as a significant difference.

### 4.3. Cell Culture and C2C12 Differentiation

The C2C12 cells were purchased from Shanghai Jianing Biotechnology Company. Resuscitated cells were cultured in dishes supplemented with 8 mL of complete medium containing 10% fetal bovine serum (FBS; Gibco, Grand Island, NY, USA). When the cell density reached 80%, digestion was carried out with trypsin (Gibco, Grand Island, NY, USA). Cells were distributed into 6-well plates and incubated for 24 h. When the cell density was approximately 50–60%, the cells were transfected into serum-free medium (Opti-MEM; Gibco, Grand Island, NY, USA) with liposomes containing vectors and incubated for additional 6–12 h. The transfection medium was then replaced with complete growth medium, and the cells were cultured for another 24 h. Then, the transfected C2C12 cells were distributed into 24-well and 96-well plates for EdU staining and CCK-8 assay, respectively. On the other hand, the transfected C2C12 cells was induced myogenic differentiation using a medium containing 2% horse serum (Gibco, Grand Island, NY, USA), and the myotube formation was observed after 6 days.

### 4.4. RNA Extraction and cDNA Synthesis

RNA extraction was performed using the Trizol reagent (Takara Bio, Kusatsu, Japan). The RNA in the nucleus and cytoplasm was extracted using a nucleocytoplasmic separation kit (Merck, Darmstadt, Germany), and the RNA concentration was measured by a nucleic acid detector. Then, cDNA synthesis was performed using the PrimeScript Reagent Kit (Takara Bio, Kusatsu, Japan), and miRNA-specific cDNA synthesis using the miRNA first strand cDNA Synthesis Kit (by stem-loop; Vazyme, Nanjing, China).

### 4.5. Vector Construction and siRNA

To overexpress circCSDE1, the full-length cDNA sequences of circCSDE1 was cloned into the pcD2.1 (GeneCreate, Wuhan, China). The siRNA for circCSDE1 (5′-ACAAUAAACAAUGAUGUUGTT-3′), miR-21-3p mimics and inhibitor were synthesized by GenePharma Company. The circCSDE1 full-length sequence with mutated miR-21-3p binding region was generated by insertion psiCHECK2 into the psiCHECK2-circCSDE1-mut vector. Mutants of *CDK16* were obtained using the same method. All vectors were verified by sequencing, and transfections were conducted with Lipofectamine 3000 reagent (Thermo Fisher Scientific, Carlsbad, CA, USA) according to the manufacturer’s instructions.

### 4.6. Quantitative Real-Time PCR

QRT-PCR assays were performed using the PerfectStart Green qRT-PCR SuperMix Kit (TransGen Biotech, Beijing, China). For circRNA and mRNA, the cDNA was synthesized using the PrimeScript^®^ RT Reagent Kit (Takara, Kusatsu, Japan); for miRNA, the cDNA was synthesized using the miRNA 1st Strand cDNA Synthesis Kit (by stem-loop) kit (Vazyme, Nanjing, China). Assays were then performed in triplicate using a SYBR Premix Ex Taq II Kit (Takara Bio, Beijing, China) and a CFX Connect Real-Time PCR Detection System (Bio-Rad, Hercules, CA, USA). Glyceraldehyde-3-phosphate dehydrogenase (*GAPDH*) was employed as the internal control for circRNA and mRNA, while U6 was used for miRNA. Primer sequences were shown in Table 1. The melting temperature(Tm) for all primers were 60 °C. The primers of circCSDE1 for qRT-PCR were divergent primers. Due to the circularization characteristics of circRNAs, it was the first to find the cyclization site of circCSDE1 (CAATG). Then, the circCSDE1 was cut and linearized between the second base (A) and the third base (A) at the cyclization site. The first 1–202 bases of the liner sequence were placed at the end to form a new sequence. The forward primer (F) and reverse primer (R) were designed on the left and right sides of the cyclization site. The final amplification product must contain the cyclization site. All of primers were designed by Primer-BLAST in NCBI (https://www.ncbi.nlm.nih.gov/tools/primer-blast/index.cgi?LINK_LOC=BlastHome (accessed on 16 August 2022)).

### 4.7. RNase R Digestion Assay

RNase R enzyme treatment group: 4 µL RNA (Longissimus dorsi of Jinfen White pig), 0.5 µL RNase R, 2 µL 10×Reaction Buffer and 13.5 µL RNase-free water. Control: 4 µL RNA (Longissimus dorsi of Jinfen White pig), 2 µL 10×Reaction Buffer and 14 µL RNase-free water. Digest in a water bath at 37 °C for 30 min, use PrimeScript Reagent Kit (Takara Bio, Kusatsu, Japan) to reverse-transcribe RNA into cDNA. Finally, the obtained cDNA was used for qRT-PCR verification.

### 4.8. EdU and CCK-8 Assays

The Kfluor555 kit (Keygen Biotech, Nanjing, China) was used for EdU staining. The CCK-8 assay was used to measure the number of cells in 96-well plates at 0 h, 12 h, 24 h, 36 h, 48 h and 60 h, with 10 µL of CCK-8 reagent added 3 h before each assay. Absorbance readings were made at 450 nm on a standard plate reader (BioTek, Winooski, VT, USA).

### 4.9. Immunofluorescence Staining

The C2C12 myoblasts treated with differentiation medium for 4 d were washed three times with PBS and fixed for 30 min with 4% paraformaldehyde. The cells were then permeabilized with 0.1% Triton X-100 for 30 min and incubated with MyHC antibody (ABclonal, Wuhan, China; 1:300) overnight at 4 °C. After removing the primary antibody, the cells were washed three times with PBS, and the fluorescent secondary antibody (1:100) was then added followed by incubation for 1 h at room temperature. Cell nuclei were stained with DAPI. Finally, the cells were washed three times with PBS and observed under a fluorescence microscope (Life Technologies, Brown Deer, WI, USA).

### 4.10. CircRNA/miRNA/mRNA Competing Endogenous RNA Network

RNAhybrid (https://bibiserv.cebitec.uni-bielefeld.de/rnahybrid (accessed on 16 August 2022)) was used to predict miRNAs likely to bind to circCSDE1, then the mRNAs predicted as targets of these miRNAs were identified using miRDB (http://mirdb.org/mirdb/comment.html (accessed on 16 August 2022)). Cytoscape software was used to display the resulting circCSDE1/miRNA/mRNA interaction network.

### 4.11. Dual-Luciferase Reporter Assays

The HEK293T cell was used for dual-luciferase reporter assays. The wild-type and mutant-type vectors of circCSDE1 were constructed. Then, three treatment groups were set up for co-transfection: (1) Co-transfected with mutant circCSDE1 and miR-21-3p mimic. (2) Co-transfected with wild type circCSDE1 and miR-21-3p mimic. (3) Co-transfected with wild type circCSDE1 and the mimic NC. The dual-luciferase reporter results for both *CDK16* and miR-21-3p were obtained in the same way. A standard plate reader (BioTek, Winooski, VT, USA) was used to measure the luciferase activity.

### 4.12. Statistical Analysis

Experimental results were expressed as mean ± SEM. All data in this study were analyzed by *t*-tests and one-way ANOVA using GraphPad Prism 5.0. Differences between means with *p* < 0.05 were considered statistically significant, and with *p* < 0.01 were considered highly statistically significant. The cycle threshold (Ct) was determined using the default threshold settings of the Mx3000p system, and the relative expression was analyzed using the 2^−ΔΔCt^ method.

## 5. Conclusions

In conclusion, we identified a novel circular non-coding RNA, circCSDE1, and validated its functions in C2C12 cells. We found that it inhibits the effects of miR-21-3p in promoting myoblast differentiation and established that it regulates growth and development of skeletal muscle through the ceRNA mechanism. This functional roles of this newly identified circCSDE1/miR-21-3p/CDK16 regulatory axis need to be analyzed in future studies.

## Figures and Tables

**Figure 1 ijms-23-12038-f001:**
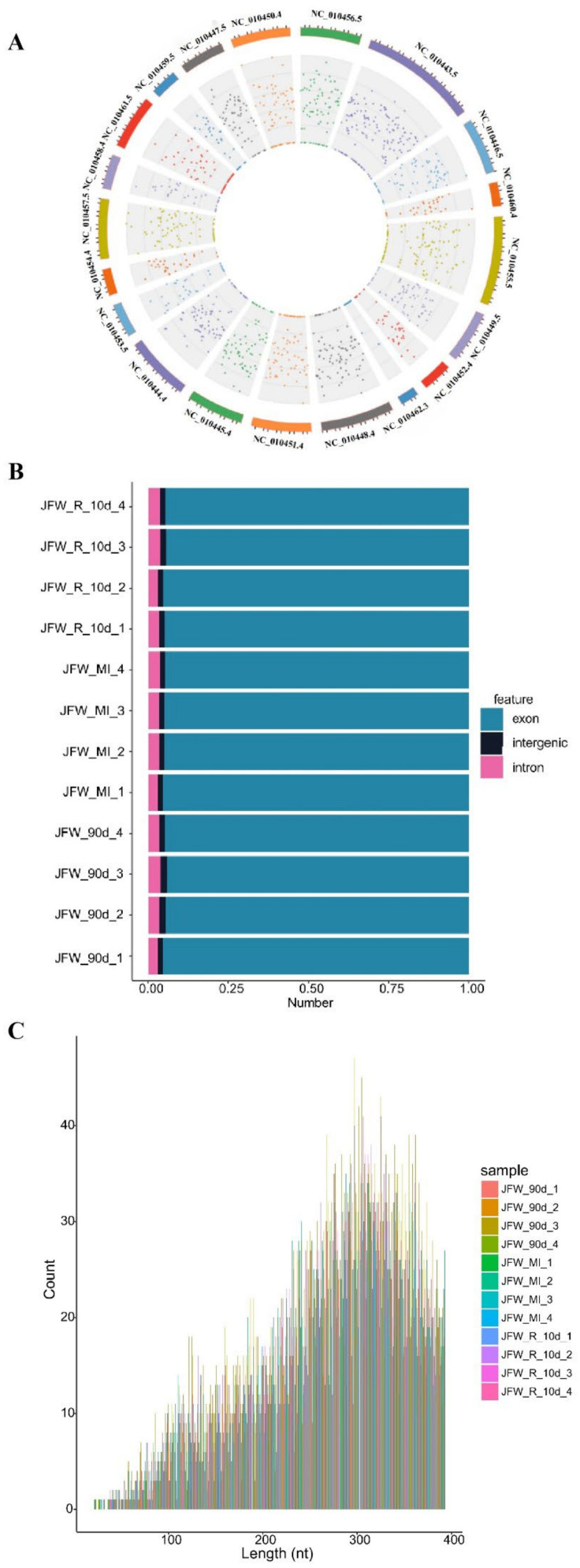
Basic profile of the candidate circRNAs obtained from sequencing data. (**A**) Circos plot showing distribution of the circRNAs across different chromosomes. (**B**) Distribution by genomic source region. (**C**) Distribution by length.

**Figure 2 ijms-23-12038-f002:**
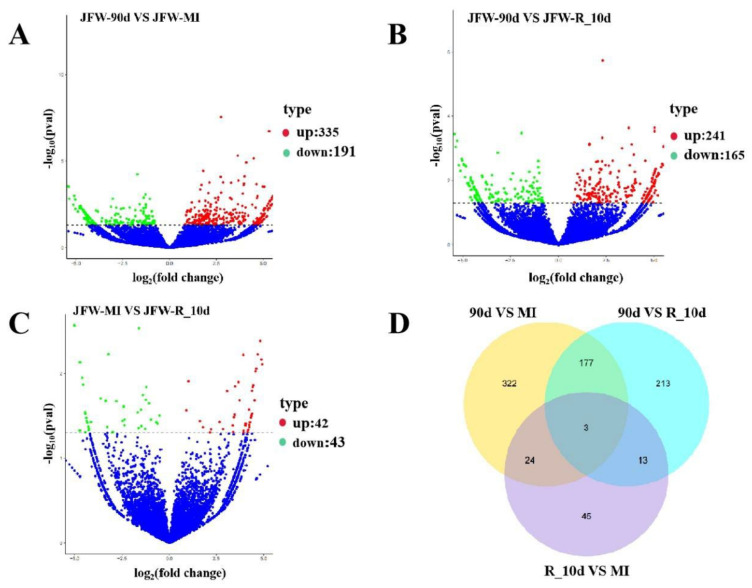
Screening for differentially expressed circRNAs. (**A**) Volcano plot of differentially expressed circRNAs in 90d and MI groups. (**B**) Volcano plot of differentially expressed circRNAs in 90d and R_10d groups. (**C**) Volcano plot of differentially expressed circRNAs in MI and R_10d groups. (**D**) Venn diagram showing overlap of differentially expressed circRNAs between groups.

**Figure 3 ijms-23-12038-f003:**
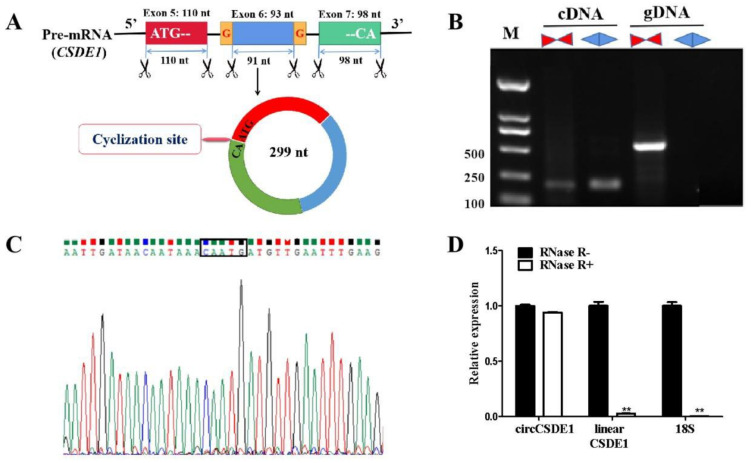
Ring-forming validation of circCSDE1. (**A**) The pre-mRNA of *C**sde1* are cyclized to form circCSDE1. (**B**) Amplified bands with convergent and divergent primers, using cDNA and gDNA as template. (**C**) The identification of circCSDE1 cyclization site sequence. (**D**) The RNase R digestion assay. **: The difference between the two groups was very significant (*p* < 0.01).

**Figure 4 ijms-23-12038-f004:**
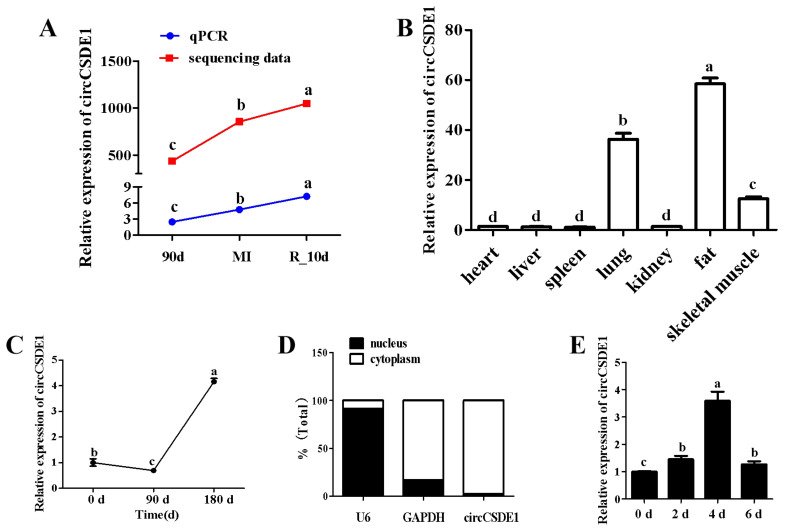
The expression patterns of circCSDE1. (**A**) The expression of circCSDE1 in 90d, MI and R_10d groups. (**B**) The expression of circCSDE1 in different tissues in Jinfen White pigs at 90 days. (**C**) The expression of circCSDE1 at different developmental stages in Jinfen White pigs. (**D**) Distribution of circCSDE1 in the nuclear and cytoplasmic fractions of myoblasts. (**E**) The expression of circCSDE1 at 0 d, 2 d, 4 d and 6 d of myogenic differentiation in C2C12 cells. “abcd”: In the same variety, the same letter indicates no significant difference, but different lowercase letters indicate significant difference (*p* < 0.05).

**Figure 5 ijms-23-12038-f005:**
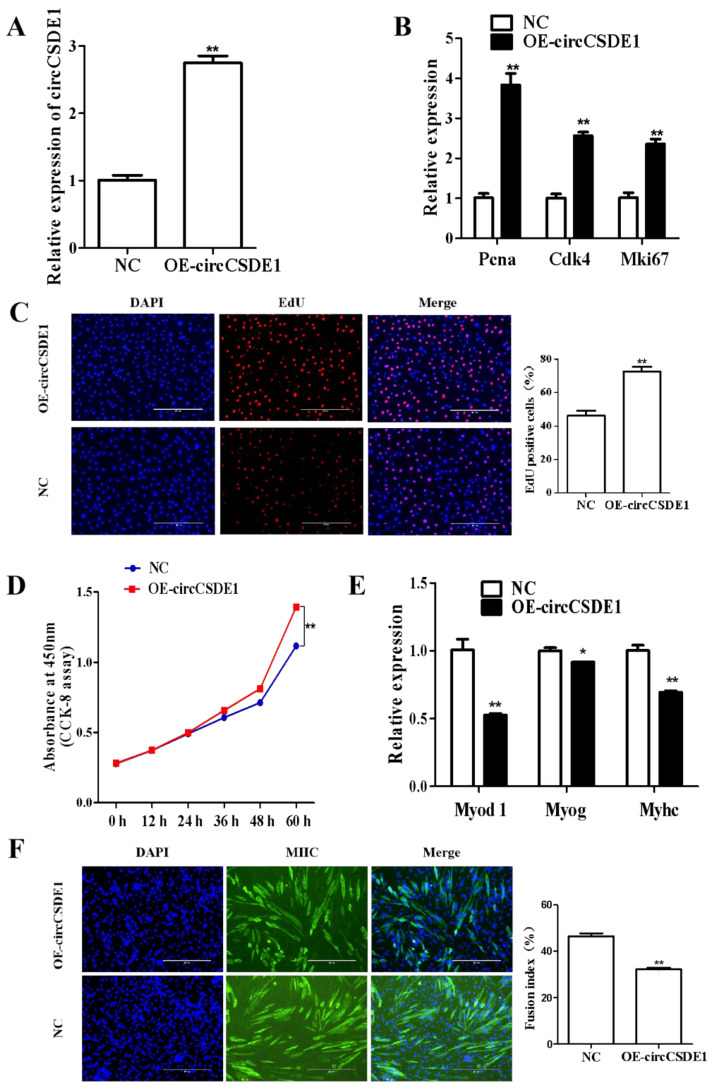
Effects of circCSDE1 overexpression on proliferation and differentiation of C2C12 cells. (**A**) Expression of circCSDE1 after transfecting the circCSDE1 overexpression vector. (**B**) Expression of proliferative genes after overexpression of circCSDE1. (**C**) EdU assay for cell proliferation, including statistics for number of positive cells. (**D**) CCK-8 assay was performed at different times. (**E**) Expression of differentiation marker genes after overexpression of circCSDE1. (**F**) Myotube formation evaluated by immunofluorescence and cell fusion index. *: The difference between the two groups was significant (*p* < 0.05); **: The difference between the two groups was very significant (*p* < 0.01).

**Figure 6 ijms-23-12038-f006:**
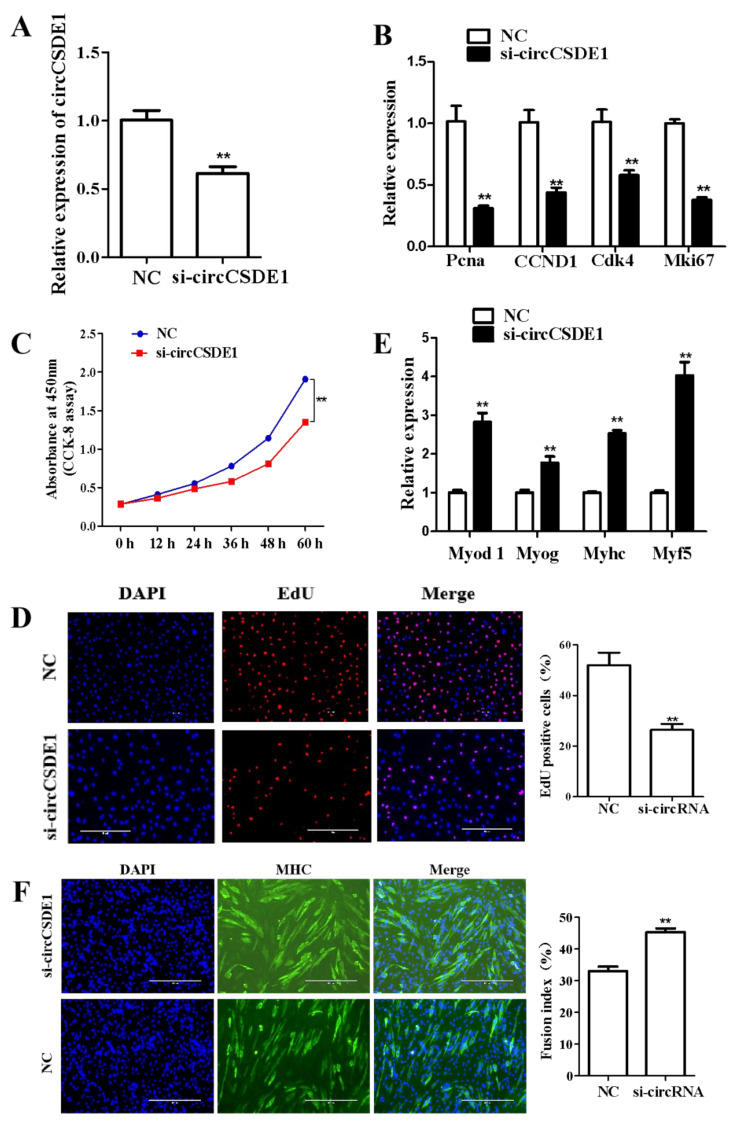
Effects of siRNA knockdown of circCSDE1 on proliferation and differentiation of C2C12 cells. (**A**) Expression level of circCSDE1 after siRNA transfection was determined by qRT-PCR. (**B**) Expression changes in proliferative genes after circCSDE1 knockdown, determined by qRT-PCR. (**C**) Cell proliferation was followed for 60 h post-transfection by CCK-8 assay. (**D**) EdU assay for cell proliferation, including statistics for number of positive cells. (**E**) Expression changes in differentiation marker genes after knockdown of circCSDE1, as determined by qRT-PCR. (**F**) Myotube formation evaluated by immunofluorescence and cell fusion index. **: *p* < 0.01.

**Figure 7 ijms-23-12038-f007:**
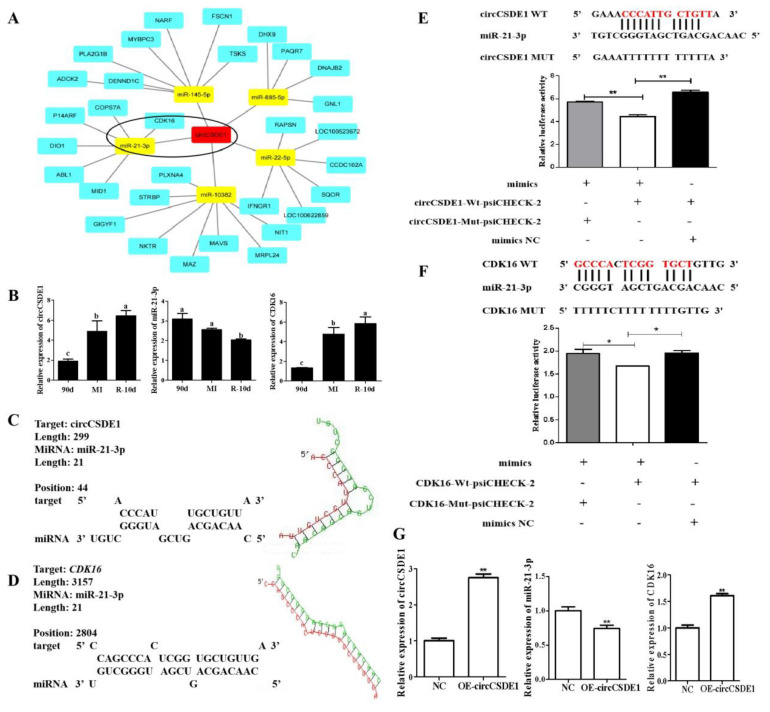
CircCSDE1 functions as a miRNA sponge in vitro. (**A**) Cytoscape network for miRNAs predicted to bind to circCSDE1. (**B**) Expression of circCSDE1, miR-21-3p, and *CDK16* in different treatment groups from the in vivo study. (**C**) RNAhybrid predictions of the binding interface between circCSDE1 and miR-21-3p. (**D**) Predicted interaction between *CDK16* and miR-21-3p, also using RNAhybrid. (**E**) Dual-luciferase reporter assay to examine binding between circCSDE1 and miR-21-3p. (**F**) Dual-luciferase reporter assay to examine binding of miR-21-3p to *CDK16*. (**G**) Changes in levels of miR-21-3p and *CDK16* in response to circCSDE1 overexpression. *: *p* < 0.05; **: *p* < 0.01. “abcd”: In the same variety, the same letter indicates no significant difference, but different lowercase letters indicate significant difference (*p* < 0.05).

**Figure 8 ijms-23-12038-f008:**
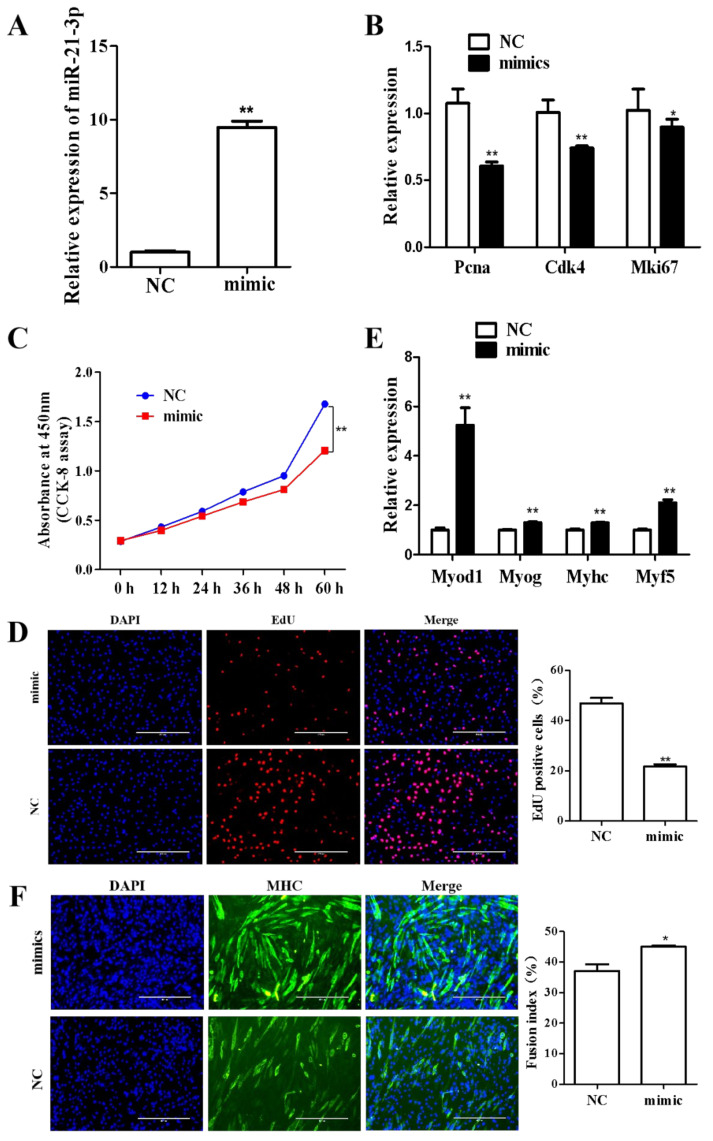
Effects of miR-21-3p overexpression on C2C12 proliferation and differentiation. (**A**) Expression level of miR-21-3p after transfection with the overexpression construct, as determined by qRT-PCR. (**B**) Expression changes in proliferative marker genes after overexpression of miR-21-3p, determined by qRT-PCR. (**C**) Cell proliferation was monitored for 60 h post-transfection by CCK-8 assay. (**D**) EdU labeling for cell proliferation, along with positive cell number statistics. (**E**) Expression changes in differentiation markers after overexpression of miR-21-3p, determined by qRT-PCR. (**F**) Myotube formation was detected by immunofluorescence and quantified using cell fusion index statistics. *: *p* < 0.05; **: *p* < 0.01.

**Figure 9 ijms-23-12038-f009:**
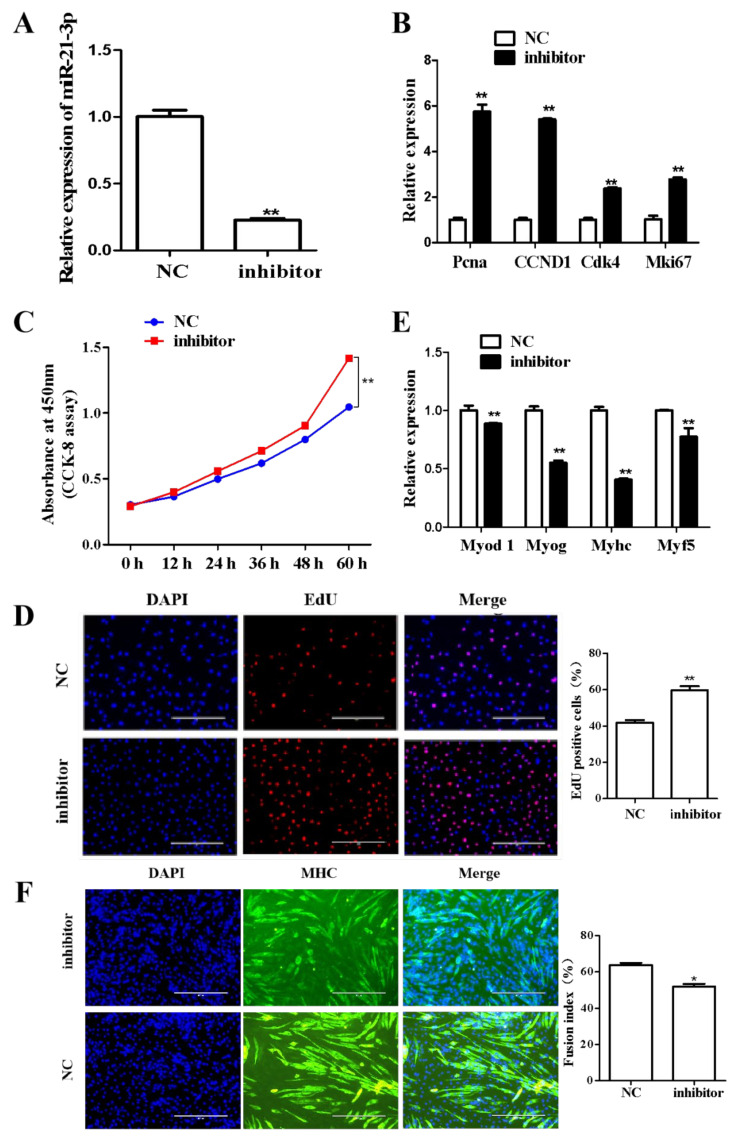
Effects of miR-21-3p knockdown on C2C12 proliferation and differentiation. (**A**) Expression level of miR-21-3p was determined by qRT-PCR following siRNA knockdown. (**B**) Expression changes in proliferative genes in response to knockdown of miR-21-3p, as determined by qRT-PCR. (**C**) Cell proliferation was followed for 60 h post-transfection by CCK-8 assay. (**D**) Proliferation was also assessed by EdU assay, and positive cell number statistics are shown. (**E**) Expression changes in differentiation marker genes after knockdown of miR-21-3p, as determined by qRT-PCR. (**F**) Myotube formation was detected by immunofluorescence and quantified using cell fusion index statistics. *: *p* < 0.05; **: *p* < 0.01.

**Figure 10 ijms-23-12038-f010:**
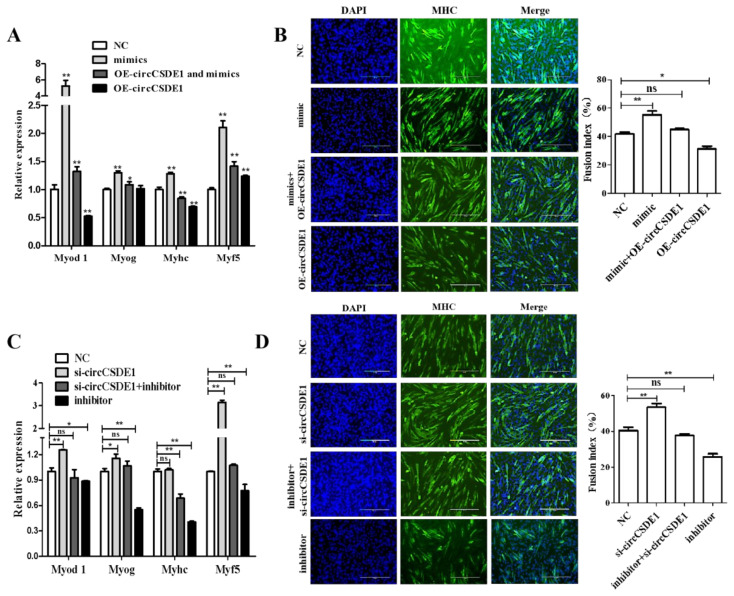
Changes in differentiation marker genes seen in rescue experiments in C2C12 myoblasts. (**A**) Expression changes of the differentiation marker genes *Myod 1*, *Myog*, *Myhc* and *Myf5* were determined by qRT-PCR following (co-)overexpression of circCSDE1 and/or miR-21-3p, as indicated in the legend. (**B**) Myotube formation was detected by immunofluorescence under the same conditions, and quantified by calculating cell fusion index. (**C**) Expression changes for the same differentiation markers after (co-)knockdown of circCSDE1 and/or miR-21-3p, as specified in the legend. (**D**) Myotube formation assay, carried out in the same way as in panel (**B**). *: *p* < 0.05; **: *p* < 0.01; “ns”: The difference between the two groups was no significant (*p* > 0.05).

**Table 1 ijms-23-12038-t001:** Gene primer sequences.

Primers	Primer Sequences (5′→3′)
circCSDE1 convergent primers	F: ACTGGGAAACCCATTGCTGT
	R: CTCCCTGTTGGACTCTGACC
circCSDE1 divergent primers	F:CCCTGAAGATGTCGAAGGGAA
	R: ATGGGTTTCCCAGTCCTACG
U6	F: CTCGCTTCGGCAGCACA
	R: AACGCTTCACGAATTTGCGT
18S rRNA	F: CCCACGGAATCGAGAAAGAG
	R: TTGACGGAAGGGCACCA
*Gapdh*	F: GCATCTTCTTGTGCAGTGCC
	R: TACGGCCAAATCCGTTCACA
*Myod1*	F:AACCATACCCCACTCTCCCC
	R:GATTTCCAACACCTGACTCGC
*Myog*	F:CAGCCCAGCGAGGGAATTTA
	R:AGAAGCTCCTGAGTTTGCCC
*Myhc*	F:TCCCAATCCCCATCGTGAAA
	R:TTTCGACTGCACCTCGTTGA
*Myf5*	F:CGGATCACGTCTACAGAGCC
	R:GCAGGAGTGATCATCGGGAG
*Mki67*	F:TAACCATCATTGACCGCTCCTT
	R:GGCCCTTGGCATACACAAAA
*Cdk4*	F:ACTCGATATGAACCCGTGGC
	R:AGCACAGACATCCATCAGCC
*Pcna*	F:CCGAGACCTTAGCCACATTG
	R:TCTCTATGGTTACCGCCTCC
*CCND1*	F:TGTGCCACAGATGTGAAGTT
	R:CAGTCCGGGTCACACTTG

## Data Availability

The original data for the RNA-seq data were submitted to the SRA Database (Submission ID: SUB11906414, BioProject ID: PRJNA869202).

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
