# Peer review of "CircCSDE1 Regulates Proliferation and Differentiation of C2C12 Myoblasts by Sponging miR-21-3p"

_ijms, 2022, doi:10.3390/ijms231912038_

Round 1

Reviewer 1 Report

Sun and colleagues developed a pig model to search for circRNAs involved with muscle growth and development. To this end, they used a whole transcriptome approach for circRNA prediction and bioinformatic analysis to identify potential regulatory networks.

A series of in vitro experiments were employed to validate the regulatory relationship between circCSDE1, miR-21-3p and gene targets.

While the approach is sound and the work is interesting, the authors did not describe how the RNA—Seq data was analyzed. Similarly, there is no information on how the circRNAs were predicted. What were the tools? The authors mentioned that circCSDE1 was a highly expressed RNA. What was the expression level? What is the significance level for selecting differentially expressed circRNAs?

Did the authors analyze the genes differentially expressed? How do they interact with the circRNAs? Or specifically with the circCSDE1?

The authors narrow down the circRNAs list to 3 targets. What is the justification for selecting only circCSDE1 for validation? How many miRNA targets were identified, and why only miR-21-3p was selected for validation? Please clarify these comments in the manuscript.

Discussion

The authors should emphasize their findings in the discussion. As of now, the discussion is too short and mainly focus on previously published work. The authors should fit their findings in the context of the mechanisms underlying myogenesis. How the repression or activation of the circCSDE1 targets affects myogenesis? Do these findings change our understanding of myogenesis? If yes, how?

The authors have the muscle transcriptome profile for these animals. I recommend integrating it to the manuscript and have a more complete picture of myogenesis in pigs.

L129-131 Com-129 pared with the age of 0 day, the expression of circCSDE1 fell slightly at the 90 days, but 130 increased about fivefold by 180 days of age (Figure 4C). In the methods section there is not mention about samples collected on day 0.

Table 1 can be included as supplementary material.

How primers for circRNA were designed? This information should be provided to the reader.

In all figures with expression levels, what is the unit for the relative expression measure? Is it ct? Please add it.

RNA-Seq data is not available on the mentioned database. Only circRNAs were uploaded.

Altogether, I recommend the authors addressing the abovementioned suggestions. The manuscript and findings are interesting. However, as presented, it has a low overall merit.

Reviewer 2 Report

Authors have performed extensive whole transcriptome sequencing analysis and identified a novel circRNA, namely circCSDE1 which has an antagonist effect on myogenic differentiation.

 However, there have some concerns that I have with this manuscript at the present stage.

Is miR-21-3p necessary in the myogenic differentiation of C2C12 cells? However, is this sufficient to inhibit cell proliferation?

Round 2

Reviewer 1 Report

The authors have addressed most of my concerns.

I have a few suggestions that I would still like to see addressed.

Please make sure all abbreviations are defined at first use.

The software find_circ and CIRI2 are both used for circRNA (L371) prediction. What was the rationale for using both software? Please comment on the feature used in each tool and the rationale. Did the authors use the default pipeline, or had specific parameters defined?

Please add the methodology used for differential expression analysis of circRNAs. It was included.

L278 – As the authors have the RNA-Seq information, did the authors investigate the expression levels of the SMAD7 gene? This information would help support the statements presented in the manuscript.

L283-284 – please explain what was meant by the reverse result.

L287-288 – What is the expression level of CSDE1 among groups? How does this relate to the expression of the candidate circRNA?

L334-337 – I would recommend the authors remove this sentence. As noted, the results presented do not support this hypothesis.

L370- Please delete “genome.”

Did the authors use any software for primer design? If yes, please add its name/reference.

A thorough English review is recommended.
